# EXAMINING THE ACHILLES' HEEL OF CLIP MODELS: THE WORST-PERFORMING CATEGORIES

## ABSTRACT

Contrastive Language-Image Pre-training (CLIP) provides a foundation model by integrating natural language into visual concepts. Although previous studies have demonstrated that satisfactory overall accuracy can be achieved across numerous downstream tasks through well-designed textual prompts, this evaluation mechanism inevitably overlooks certain categories because the impact of some underperforming categories on overall performance remains limited, even if they are highly important. For example, on ImageNet, there are a total of 10 categories with class-wise accuracy as low as 0%, which is significantly inferior to the overall performance of 64.1%. This phenomenon reveals the potential risks of using CLIP models, especially in risk-sensitive applications. To address this issue, we investigate the alignment between the two modalities in the CLIP model and propose the Class-wise Matching Margin (CMM) to measure the inference confusion. CMM can effectively identify the worst-performing categories and estimate the potential performance of the candidate prompts. We further query large language models to enrich descriptions of worst-performing categories and build a weighted ensemble to highlight the efficient prompts. Experimental results clearly verify the effectiveness of our proposal, where the accuracy on the worst-10 categories on ImageNet is boosted to 5.2%, without manual prompt engineering, laborious optimization, or access to labeled validation data.

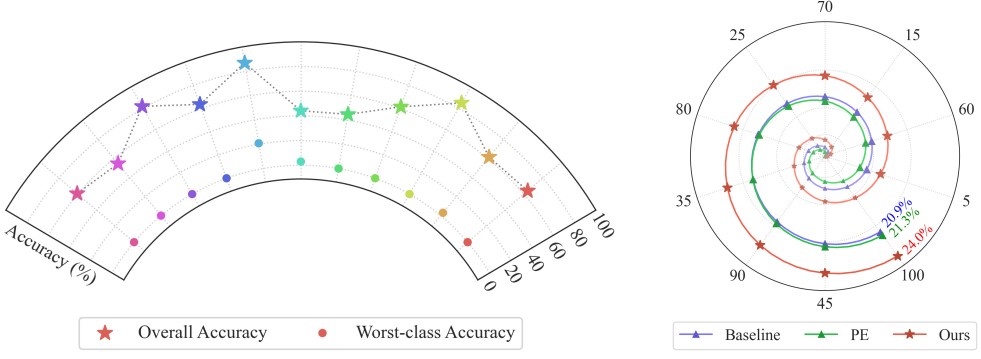

Figure 1: (**L**) Overall vs. Worst-class Acc. across 11 benchmarks. (**R**) Worst-100 Acc. on ImageNet.

# 1 INTRODUCTION

Recent advances in language-guided visual pre-training models have achieved great success, such as CLIP (Radford et al., 2021), ALIGN (Jia et al., 2021), and ImageBind (Girdhar et al., 2023), showing strong generalization on a variety of downstream tasks, including image classification, object detection (Shi et al., 2022), image generation (Nichol et al., 2022), etc. By establishing a connection between visual concepts and natural language, the vision-language models could utilize the textual description for target categories to implement image classification, even without further learning on the specific training data. Given a specific task, CLIP is utilized to implement image classification using cross-modal matching. The developers could design some textual prompts, such as "a photo of

a [CLASS]", where [CLASS] can be substituted with any relevant description, such as the class name from a vocabulary. The prediction is then provided by identifying the prompt-formed class description that has the highest similarity to the input image. Zero-shot CLIP is effective and can even match the performance of standard classification models that have access to training examples (Radford et al., 2021): CLIP with a ViT-L/14 vision encoder matches the ImageNet accuracy of a ResNet-101. Through prompt engineering, the zero-shot performance of the CLIP classifier has been continuously improved. For example, the CLIP with ViT-B/16 has increased its overall accuracy on ImageNet from 64.18% to 66.92% and 68.57% when using the prompt "A photo of a [CLASS]" and an ensemble of 80 hand-crafted templates, rather than just the "[CLASS NAME]". Unfortunately, we find that behind this series of successes, their worst-5 performing categories consistently show 0 class-wise accuracy. The left part of Figure 1 provides a further comparison across 11 classification benchmarks and indicates that the performance of individual categories is significantly inferior to the overall accuracy. This phenomenon clearly reveals the limitations of previous evaluations of CLIP with regard to its overall performance, that is, leaving actual performance on individual categories in the shadow.

In this work, we investigate the cross-modal alignment of CLIP and provide an estimate for class-wise performance, i.e., Class-wise Matching Margin (CMM). CMM measures the margin of similarity between image representations and different prompts, capturing the degree of class-wise confusion. Taking the CMM, we could effectively identify the worst-performing categories and further estimate the potential performance of the candidate prompts. For the identified categories with weak cross-modal alignment, we query large language models to enrich their category-wise description text. For the identified prompts with better CMM, we strengthen them and finally construct a zero-shot prompt ensemble method CPE (CMM-based Prompt Ensemble). Experimental results on 12 benchmarks clearly verify the effectiveness of our proposal. Additionally, the accuracy on the worst-performing categories has been consistently improved, as shown in the right part of Figure 1. The performance of the worst-10 categories on ImageNet has been improved from 0% to 5.2%, and the worst-100 accuracy on ImageNet has been boosted to 24.0%.

## 2 ZERO-SHOT PREDICTION OF CLIP

### 2.1 PRELIMINARIES

CLIP has shown a strong generalization capability on zero-shot predictions across various downstream tasks (Radford et al., 2021). This is attributed to its pre-training on a web-scale dataset (Schuhmann et al., 2022), wherein CLIP aligns the representation between pairs of retrieved images and texts, by maximizing the cosine similarity. For a specific downstream task, with provided textual prompts, it utilizes cross-modal matching to implement image recognition, without the requirement for a subsequent training process. For a specific classification task containing $N$ categories, CLIP first constructs $N$ corresponding textual prompts $P = \{\text{prompt}_i\}_{i=1}^N$. The $\text{prompt}_i$ could be any natural language description for the class $i$, such as "a photo of a dog" or "a tuxedo cat with a long tail". During the inference stage, CLIP categorizes an image by matching the most similar prompt. Formally, given an image $x$, the predicted label can be calculated by

$$\hat{y} = \arg\max_{i \in [N]} \ \text{sim}(x, \text{prompt}_i), \tag{1}$$

where $\text{sim}(x, \text{prompt}_i)$ is defined as the cosine similarity of the embedding of $x$ and $\text{prompt}_i$, i.e., $\text{sim}(x, \text{prompt}_i) = \frac{f_I(x)}{\|f_I(x)\|} \cdot \frac{f_T(\text{prompt}_i)}{\|f_T(\text{prompt}_i)\|}$, given that $f_I$ and $f_T$ denote the image and text encoder of CLIP. In this way, the image classification task is reformulated to an image-text matching problem.

Proper construction and utilization of prompts is critical to the matching performance. A widely-used default template of CLIP is "a photo of a [CLASS]", which has been found to provide stable and good zero-shot performance. Prompt engineering and prompt ensemble have been found to further improve performance. For instance, a customized template "a satellite photo of a [CLASS]" for a satellite image classification task, or ensembles the prediction results calculated by multiple templates, such as "a photo of a big [CLASS]" and "a photo of a small [CLASS]". The most relevant work to us is ZPE (Allingham et al., 2023), which provides a method for building a zero-shot ensemble based on normalized confidence to assign weights to prompt templates. Even though the overall accuracy of CLIP has been further improved through ZPE, the examination on underperforming categories is still unexplored.

## 2.2 MEASURING UNDERPERFORMING CATEGORIES

Although previous studies have demonstrated overall accuracy being continuously improved across numerous downstream tasks through well-designed text prompts, this evaluation mechanism inevitably overlooks certain categories because the impact of some underperforming categories on overall performance remains limited, even if they are highly important. In this paper, we emphasize the need to focus on the performance of individual categories, especially those that continue to demonstrate unsatisfactory performance as the worst-performing categories. The class-wise accuracy could be formulated as:

$$\text{Acc}_i = \frac{1}{|C_i|} \sum_{(x,y) \in C_i} \mathbb{I}(y = \hat{y}), \tag{2}$$

where $C_i$ denotes the set of samples with ground-truth label $y = i$. Existing works mainly evaluate CLIP with overall accuracy on the entire test set, which is approximate to calculate the mean value of $\{\text{Acc}_i\}_{i=1}^N$. This inevitably neglects the worst-performing categories, which do not have much consequence even if they register zero accuracy.

The worst-class accuracy $\min_{i \in [N]} \text{Acc}_i$ could address it which focuses on the poorest category. To include more underperforming categories in the evaluation and provide a comprehensive perspective, we extend it to the worst-$k$ class accuracy, which calculates the average of the $k$ smallest $\text{Acc}_i$:

$$\text{Worst@}k = \frac{1}{k} \min_{K \subseteq [N], |K| = k} \sum_{i \in K} \text{Acc}_i. \tag{3}$$

Apart from the worst-class accuracy, we additionally introduce two metrics that characterize the performance distribution across all categories: harmonic mean and geometric mean accuracy. These metrics will be employed in the experiments and discussed in Section 6.1. These metrics emphasize the focus on categories of performance that are not yet satisfactory, equipping us with effective tools to investigate the limitations of existing CLIP applications.

## 3 CLASS-WISE MATCHING MARGIN

In this section, we first investigate the scheme of cross-modal matching between image and textual prompts and propose the Class-wise Matching Margin (CMM) as a measure to quantify the degree of confusion between categories. CMM not only provides a measure to identify the worst-performing categories but also can be further used to evaluate the quality of different prompt templates, providing a promising way to boost the worst-performing categories.

### 3.1 DEFINITION OF CMM

The engine of CLIP is the cross-modal similarity matching between images and textual prompts. Considering an image $x$ and its grounding label $i$, the misclassification of CLIP occurs only when there exists another class $j$ with higher similarity to image $x$, $\text{sim}(x, \text{prompt}_j) > \text{sim}(x, \text{prompt}_i)$. Otherwise, the image $x$ will be correctly classified into category $i$. Given more images $x \sim C_i$, we could have a similarity measure $H$ to evaluate the matching degree between visual categories and textual prompts. It is formally defined as:

$$H_{ij} := \mathbb{E}_{x \sim C_i}[\text{sim}(x, \text{prompt}_j)] \tag{4}$$

In the above definition, $C_i$ represents the distribution of images possessing the label $i$, i.e., $C_i = P(x|y = i)$. Similar to our previous analysis on the instance level, if we have $H_{ij} > H_{ii}$, then it is expected that images belonging to class $i$ are more likely to be misclassified as class $j$. Given a specific image dataset, we could derive an empirical estimation $\hat{H}$:

$$\hat{H}_{ij} = \frac{1}{|C_i|} \sum_{x \in C_i} \text{sim}(x, \text{prompt}_j), \tag{5}$$

where $C_i$ represents the set of samples possessing the label $i$. A smaller margin between $H_{ii}$ and $H_{ij}$ means that the images assigned to $C_i$ have a high similarity to prompt $j$, corresponding to a higher

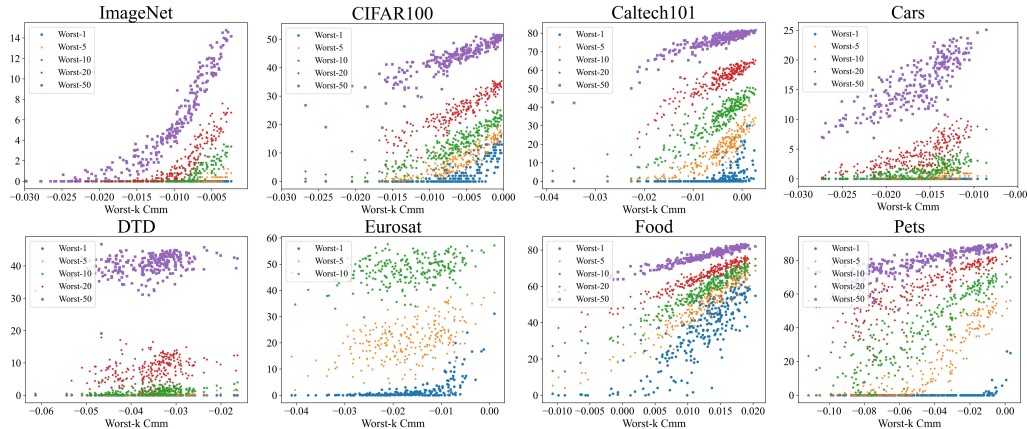

Figure 2: Performance of different textual templates on first 8 datasets with the ViT-B/16 backbone. Each scatter represents a specific textual template. The value of Worst-$k$ CMM ($k = 0.1N$) is approximately in direct proportion to the worst-category performances.

degree of confusion. Based on this connection, we present the Class-wise Matching Margin (CMM) as:

$$\text{CMM}_i = \hat{H}_{ii} - \max_{j \neq i} \hat{H}_{ij} \tag{6}$$

CMM considers the margin between the similarity of images to their corresponding textual prompt and the similarity to other prompts that are most susceptible to confusion. Consequently, when a category has a lower CMM value, its class-wise accuracy is expected to be unsatisfactory. This provides an approach to identifying the worst-performing categories. Unlike class-wise accuracy which solely determines 0-1 correctness, CMM allows for a more precise expression of the degree to which worst-performing categories are confused. Consequently, this facilitates the targeted design of algorithms to boost the worst-performing categories of the CLIP model.

### 3.2 CMM ASSESSING PROMPT TEMPLATES

In the previous section, we propose CMM and analyze that it can assist in identifying the worst-performing categories. In this section, we further discovered that the capability of CMM can be extended to evaluate the worst-k performance of prompt templates, thus serving as a basis for quality assessment. As discussed above, templates are the basis of prompts across different classes, as a prompt is a combination of the template and a specific [class]. Therefore, evaluating the quality of templates is a key step in implementing an effective prompt ensemble. Similar to worst-k accuracy, we use the averaged CMM score of the worst $k$ classes, worst-$k$ CMM

$$\text{Worst-}k \text{ CMM} = \frac{1}{k} \min_{|K|=k} \sum_{i \in K} \text{CMM}_i \tag{7}$$

In the context of assessing accuracy at the class level, we argue that Worst-$k$ CMM can also be used to evaluate the worst-class performance across different prompt templates. To verify this perspective, we conducted experiments on 8 datasets with more than 200 candidate templates and visual a detailed connection between Worst-$k$ CMM and the performances of the corresponding worst categories. The results on more datasets and the details of the templates are available in the appendix. As shown in Figure 2, it could be found that the Worst-$k$ CMM is roughly proportional to the accuracy of the worst-performing categories across benchmark datasets, including a series of different domains. This result indicates that we can assess the quality of different textual templates by using the Worst-$k$ CMM estimation.

### 4 THE PROPOSED CPE

In the section, we present our method CPE by utilizing CMM to build a zero-shot Prompt Ensemble serving the worst-performing categories. CPE improves the worst-performing categories from two

aspects: prompt templates and category descriptions. Firstly, it uses CMM identity to identify categories that are prone to confusion and enhances their textual descriptions. Then, it assigns weights to different templates using CMM to construct an ensemble.

## 4.1 ENRICHING THE DESCRIPTIONS VIA LARGE LANGUAGE MODEL

Taking the benefits of CMM, we could effectively identify the worst-performing categories and then enhance them. Following the (Menon & Vondrick, 2023), we automatically construct a description for these categories from a large language model, such as GPT-3, to provide a detailed class-wise description and boost the textual-image matching. We take the QA-based prompt as the input of the language model to generate the detailed descriptions, as shown in Figure 3 The specific class-dependent token, *but-*

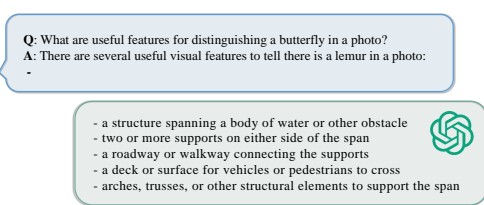

Figure 3: Enriching the description from GPT-3.

*terfly*, could be substituted for any identified category that needs to be enhanced. Next, the diverse textual attributes provided by the language model could help CLIP better distinguish visual features in images. Note that, while large language models do not directly access images for specific tasks, their training data includes textual descriptions from the real world. This provides them with powerful external knowledge to enhance the specified categories. Given the multiple descriptions for class $i$, we calculate an averaged similarity for them as a replacement for the raw $\text{sim}(x, \text{prompt}_i)$ in Equation 1.

## 4.2 WEIGHTED PROMPT ENSEMBLE

Based on CMM's ability to evaluate the performance of different templates, we develop a weighted prompt ensemble method based on Worst-$k\%$ CMM. Given a candidate pool $P$ of $m$ templates: $P = \{\text{template}_i\}, i = 1, 2, .., m$, where each template will be assigned a weight, denoted as $w_i$. The final prediction for class $j$ can be calculated by the weighted ensemble:

$$\hat{y} = \arg \max_{j \in [N]} \sum_{i=1}^{p} \text{sim}(x, \text{prompt}_{ij}) \cdot w_i, \tag{8}$$

where $\text{prompt}_{ij} = \text{template}_i \cdot \text{class}_j$ is defined as the concatenation of the template and the label. The weight $w_i$ is obtained by the Worst-$k$ CMM evaluation on each template after softmax normalization.

$$w_i = \frac{\exp\left(\text{Worst-}k \text{ CMM}\left[\text{template}_i\right]\right)}{\sum_{j=1}^{m} \exp\left(\text{Worst-}k \text{ CMM}\left[\text{template}_j\right]\right)}, \quad i = 1, 2, \ldots, m \tag{9}$$

We observe that certain templates exhibit a significant bias towards particular datasets. For example, the template '*satellite photo of a* {}.' demonstrates such bias in the case of ImageNet due to the infrequent occurrence of satellite photos within this dataset. As a result, ensembling these biased templates into the final predictions can yield unfavorable outcomes. Motivated by this observation, we employ the prompt selection technique to selectively include templates with high weights while disregarding those with low weights. Concretely speaking, we set a threshold $\tau$ to judge which template should be selected. We simply set it as the median of weights to avoid the laborious effort of adjusting hyper-parameters. The final prediction is rectified to:

$$\hat{y} = \arg \max_{j \in [N]} \sum_{i=1}^{p} w_i \mathbb{I}(w_i \geq \tau) \text{sim}(x, \text{prompt}_{ij}) \tag{10}$$

The CMM provides a measure to identify the worst categories and assess the quality of candidate templates. However, the ground truth $C_i$ in Equation 5 is not available in practice. We directly **use the pseudo label as the replacement to calculate CMM**, then enrich the descriptions of the worst K classes (Section 4.1) and assign weights to templates (Section 8), **without any usage of ground-truth labels**. All the following experiments are conducted by the pseudo CMM.

## 5 RELATED WORK

**Contrastive Language-Image Pre-training (CLIP).** CLIP aligns the visual concepts and natural language via contrastive learning on the 400 million curated image-text pairs (Radford et al., 2021). Through template-form textual prompts, CLIP could implement image classification in a zero-shot manner, which has become a prominent paradigm for visual recognition. Zero-shot CLIP is effective and can even match the performance of classification models that have access to training examples. The subsequent work suggests that customizing the templates for specific tasks (Prompt Engineering) or combining the prediction results calculated by different prompts (Prompt Ensemble) can further explore the capabilities of CLIP in downstream tasks (Wang et al., 2023a; Menon & Vondrick, 2023). As a result, in the recently proposed ZPE (Allingham et al., 2023), a zero-shot prompt ensemble method, the zero-shot performance on ImageNet could be improved to 68% using the ViT-B/16 backbone. However, we regretfully found that the performance in certain categories has been overlooked, which has raised our serious concerns regarding the practicality of CLIP. In this work, we call for attention to the worst-performing categories of CLIP, in order to enhance its practicality.

**Remedy the underperforming categories**. Regarding the underperforming categories, there are also some exploration and solutions in the machine learning community. One perspective is to study the impact of hard classes or examples (Shrivastava et al., 2016; Suh et al., 2019; Xuan et al., 2020), and propose Hard Example Mining methods to alleviate this issue. Another perspective is to delve into the inherent challenge in the task, such as the imbalanced class distribution and the data scarcity issues, then utilize the Robust Optimization approaches to encourage more robust worst-case performance (Chen et al., 2017; Samuel & Chechik, 2021; Wang et al., 2023b; Du & Wu, 2023). However, these works focus on remedying underperforming classes during the training process, which is different from the zero-shot recognition considered in this work.

## 6 EXPERIMENTS

To validate the effectiveness of our method, we have conducted experiments on ImageNet and 11 widely used benchmarks from various domains. We first introduce the benchmark datasets and the evaluation metrics. Then we present the comparison of our method with the competing baselines. Finally, we provide a detailed analysis to promote a deeper understanding of CPE.

Note that all of the following experiments are conducted based on pseudo-CMM, which utilizes pseudo-labeling to calculate Worst$-k$ CMM without the need for ground-truth labels. For different datasets, we directly use the template pool provided in the original CLIP paper as the candidate prompts for all methods. These candidate prompts are also provided in the appendix for completeness. For the choice of $k$, we default it to 10% of the number of classes in the dataset to accommodate the needs of datasets with different scales.

### 6.1 EXPERIMENTAL SETTINGS

**Datasets.** We conduct experiments on ImageNet (Deng et al., 2009) and 11 additional benchmarks: CIFAR100 (Krizhevsky et al., 2009), Caltech101 (Fei-Fei et al., 2004), Cars196 (Krause et al., 2013), DTD (Cimpoi et al., 2014), EuroSat (Helber et al., 2019), Food-101 (Kaur et al., 2017), Oxford-Flowers (Nilsback & Zisserman, 2008), Oxford-Pets (Parkhi et al., 2012), Resisc45 (Cheng et al., 2017), SUN397 (Xiao et al., 2010), and FGVCAircraft (Maji et al., 2013), covering a series of specific domains, as well as fine-grained and specialized tasks.

**Evaluation metrics.** To examine CLIP w.r.t. the under-performing categories, we mainly use the worst-$k$ accuracy to evaluate the predicted results. The definition can be referred to in Section 2.2. The overall accuracy is also reported for thorough comparison. Moreover, to characterize the performance distribution across all categories, we calculate the harmonic mean (HM) and the geometric mean (GM) accuracy.

$$\text{HM}(\{\text{Acc}_i\}_{i=1}^N) = \frac{N}{\frac{1}{\text{Acc}_1} + \cdots + \frac{1}{\text{Acc}_N}}, \quad \text{GM}(\{\text{Acc}_i\}_{i=1}^N) = \sqrt[N]{\text{Acc}_1 \cdot \cdots \cdot \text{Acc}_N}$$

Table 1: Zero-shot prediction accuracy (%) on ImageNet dataset. Both worst@$k$ (simplified as @$k$) and overall accuracy are reported. **Bold** numbers represent the best results across all methods; underline numbers represent the second best results. †: without category discriptions; ‡: without prompt selection; *: ZPE accesses to the pre-training dataset to help downstream tasks.

| Methods | CLIP ViT-B/16 | | | | | | CLIP ViT-L/14 | | | | | |
|---|---|---|---|---|---|---|---|---|---|---|---|---|
| | @5 | @10 | @20 | @50 | @100 | Overall | @5 | @10 | @20 | @50 | @100 | Overall |
| Class Name | 0.00 | 0.00 | 1.40 | 7.20 | 14.70 | 64.06 | 0.00 | 0.60 | 3.10 | 11.20 | 21.26 | 71.57 |
| 'A photo of a {}.' | 0.00 | 3.20 | 6.40 | 13.28 | 20.88 | 66.75 | 0.40 | 2.80 | 7.80 | 17.36 | 26.42 | 73.47 |
| Descriptor | 0.80 | 2.60 | 7.10 | 14.84 | 22.20 | 67.88 | 2.40 | 4.00 | 9.50 | 19.80 | 29.04 | 74.86 |
| Prompt Ens. | 0.00 | 1.20 | 4.50 | 11.96 | 21.32 | 68.22 | 0.80 | 3.20 | 7.90 | 19.52 | 29.64 | 75.35 |
| MLS | 0.00 | 1.40 | 4.60 | 12.04 | 21.44 | 68.25 | 1.20 | 3.40 | 8.00 | 19.60 | 29.76 | 75.35 |
| ZPE* | 0.00 | 1.40 | 4.80 | 12.12 | 21.46 | **68.27** | 1.20 | 3.40 | 8.10 | 19.68 | 29.78 | **75.37** |
| CPE† | 0.40 | 1.60 | 4.70 | 12.00 | 21.46 | 68.26 | 1.20 | 3.40 | 8.10 | 19.72 | 29.76 | **75.37** |
| CPE‡ | **3.60** | 5.00 | 7.90 | **15.68** | **24.18** | 67.91 | **3.20** | 5.20 | 8.90 | 19.40 | 30.00 | 75.11 |
| CPE | 2.80 | **5.20** | **8.70** | 15.48 | 23.96 | 67.66 | 2.00 | **5.20** | **10.00** | **20.00** | **30.28** | 74.99 |

Table 2: Zero-shot Performance (%) on 11 datasets. We calculate the worst@$k$, harmonic mean (HM), geometric mean (GM), and overall accuracy, and report the average results of all datasets. **Bold** and underline numbers represent the best and the second best results across all methods, respectively. †: without category discriptions; ‡: without prompt selection; *: ZPE requires the pre-training data.

| Methods | CLIP ViT-B/16 | | | | | | | |
|---|---|---|---|---|---|---|---|---|
| | Worst@1 | Worst@5 | Worst@10 | Worst@20 | Worst@50 | HM | GM | Overall |
| Class Name | 2.44 | 10.43 | 19.91 | 28.40 | 44.58 | 12.08 | 14.97 | 60.53 |
| 'A photo of a {}.' | 5.84 | 15.44 | 23.32 | 31.52 | 47.81 | 13.10 | 15.14 | 63.29 |
| Descriptor | 7.66 | 16.20 | 23.56 | 31.58 | 48.15 | 25.98 | 31.65 | 64.19 |
| Prompt Ens. | 10.06 | 18.31 | 26.36 | 34.52 | 50.84 | 30.96 | 33.52 | 65.74 |
| MLS | 10.04 | 18.33 | 26.32 | 34.51 | 50.86 | 30.96 | 33.51 | 65.74 |
| ZPE* | 9.96 | 18.33 | 26.27 | 34.48 | 50.85 | 30.96 | 33.53 | 65.75 |
| CPE† | 10.57 | 18.69 | 26.50 | 34.70 | **51.05** | 31.33 | 33.73 | **65.90** |
| CPE‡ | 12.08 | 18.75 | 27.28 | 35.09 | 50.60 | 31.21 | 33.81 | 65.83 |
| CPE | **12.46** | **19.47** | **27.53** | **35.11** | 50.71 | **31.70** | **34.01** | 65.86 |
| Methods | CLIP ViT-L/14 | | | | | | | |
| | Worst@1 | Worst@5 | Worst@10 | Worst@20 | Worst@50 | HM | GM | Overall |
| Class Name | 6.53 | 16.82 | 24.83 | 34.44 | 51.60 | 16.27 | 18.98 | 65.78 |
| 'A photo of a {}.' | 13.31 | 24.55 | 32.77 | 41.08 | 56.55 | 23.36 | 23.66 | 70.11 |
| Descriptor | 13.42 | 22.28 | 30.07 | 39.04 | 56.24 | 32.84 | 35.53 | 70.52 |
| Prompt Ens. | 15.25 | 27.37 | 35.60 | 44.48 | 59.57 | 41.69 | 48.76 | 72.49 |
| MLS | 15.28 | 27.44 | 35.60 | 44.46 | 59.59 | 41.64 | 48.76 | 72.50 |
| ZPE* | 15.51 | 27.39 | 35.58 | 44.50 | 59.60 | 41.74 | 48.78 | 72.52 |
| CPE† | 15.44 | 27.37 | 35.59 | 44.52 | **59.64** | **41.79** | **48.86** | 72.56 |
| CPE‡ | **17.75** | **27.92** | **35.99** | **44.58** | 59.45 | 41.72 | 48.81 | **72.78** |
| CPE | 15.59 | 27.28 | 35.87 | 44.55 | 59.45 | 39.98 | 48.54 | 72.75 |

## 6.2 MAIN RESULTS

**Results on ImageNet.** In Table 1, we present the worst 5 to 100 class accuracy, along with the overall accuracy on ImageNet. The CLIP baseline, which directly uses class names for prompts, achieves an overall accuracy of 64.06%. However, it performs significantly worse on individual categories compared to the overall metric, where there are 10 categories even showing a class-wise accuracy of 0. Appropriate prompt templates and Prompt Ensemble have been found to improve the overall accuracy of CLIP in previous work. The recently proposed ZPE, an effective weighted ensemble method (Allingham et al., 2023), takes the lead in terms of overall accuracy. However, we can find that its improvement in worst@$k$ performance is very limited, revealing the potential risks of existing efforts, as discussed in the above section. Descriptor (Menon & Vondrick, 2023) enhances the textual descriptions of categories to improve the image recognition capability of CLIP. This approach aligns with our strategy of improving the performance of the worst-performing categories, but it still shows limited effectiveness. One possible reason is that enhancing descriptions for all categories without differentiation may suppress the performance of the worst-performing categories,

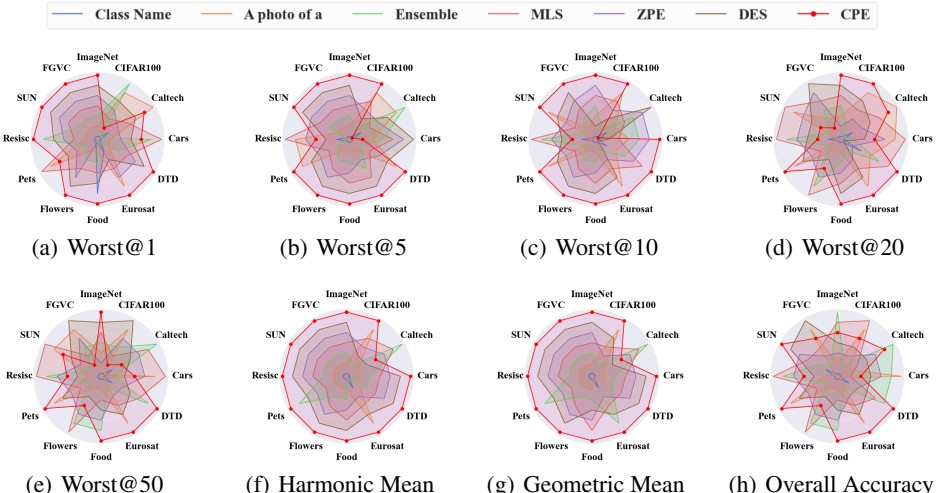

Figure 4: Results with ViT-B/16 Backbone across varying metrics.

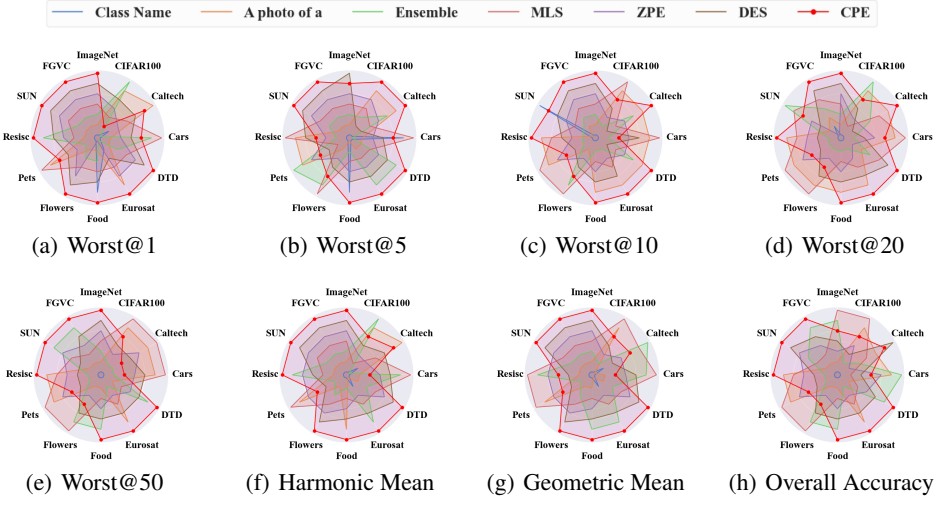

Figure 5: Results with ViT-L/14 Backbone across varying metrics.

making it challenging to improve their performance. Additionally, CPE has demonstrated comparable performance to existing methods in terms of overall accuracy, even though we did not optimize it. This may suggest that overall performance and worst-class performance can be achieved simultaneously without sacrificing either, showing a promising way to extend the practicality of CLIP.

**Results on the fine-grained and specialized benchmarks.** Apart from ImageNet, we conduct experiments on the 11 fine-grained and specialized classification benchmarks. Due to the space limitation, we report the average results in Table 2 and leave the detailed values in the appendix. These 11 datasets are from various fields with each containing approximately 50 to 400 categories. From the results, we could find that CPE consistently achieves the best performance. Specifically, it increases the worst@1 accuracy by more than 2%. Also, the HM, GM, and overall accuracy of CPE surpass the previous methods, clearly verifying the effectiveness of CPE in improving both worst-performing categories. Figure 4 and Figure 5 show the rankings of different methods for the given metric across 12 datasets, to further examine their generalization. The results show that CPE achieves the best results in most of the datasets, especially regarding the worst@1, worst@5, and worst@10 accuracy. This demonstrates that CPE exhibits a significant effect on improving the

performance of the worst categories. The excellent performance across benchmarks demonstrates the advantage of CPE in generalization ability, highlighting its practicality.

## 6.3 Additional Analysis

**Selection on unified template pool.** In CPE, we follow previous works (Radford et al., 2021; Allingham et al., 2023) to use a handcraft template pool for each dataset. One may be concerned with using a unified and larger template pool. In this paragraph, we integrate the templates from different datasets to construct a larger pool to examine the ability to eliminate poor templates (Allingham et al., 2023). The comparison of different ensemble methods is reported in Table 3. For simplification, averaged prompt ensemble baseline and max-logits scoring are denoted as PE and MLS, respectively. The best results are bolded. Compared with handcraft templates in Table 1 and 2, the results have deteriorated

Table 3: Performance comparison of prompt ensemble methods on ViT-B/16 using a unified template pool.

| Methods | ImageNet | | | | | |
|---|---|---|---|---|---|---|
| | @5 | @10 | @20 | @50 | @100 | Overall |
| PE | 0.00 | 0.40 | 2.20 | 8.76 | 18.0 | 67.6 |
| MLS | 0.00 | 0.40 | 2.30 | 8.88 | 18.1 | 67.6 |
| ZPE* | 0.00 | 0.40 | 2.70 | 9.44 | 18.6 | **67.8** |
| CPE | **1.20** | **3.20** | **5.60** | **11.6** | **19.9** | 67.6 |
| Methods | 11 fine-grained dataset | | | | | |
| | @1 | @5 | @10 | @20 | @50 | Overall |
| PE | 4.97 | 11.4 | 22.0 | 31.2 | 47.7 | 63.6 |
| MLS | 5.02 | 11.6 | 22.1 | 31.3 | 47.8 | 63.6 |
| ZPE* | 5.35 | 12.2 | 22.7 | 31.7 | 48.3 | 63.9 |
| CPE | **6.19** | **15.8** | **25.1** | **33.3** | **49.3** | **64.6** |

because the candidate pool includes templates from different domains, which may not be suitable for a certain task. Nevertheless, CPE consistently achieves the highest worst-class accuracy compared to existing ensemble methods. This demonstrates that CPE could efficiently strengthen the useful template from a lower-quality candidate pool and is less sensitive to the different templates.

**Impact of descriptions augmentation** In this work, we propose to enrich the category descriptions for the worst-performing categories. The most related work descriptor (Menon & Vondrick, 2023) performs augmentation on all categories to enhance overall accuracy. In this paragraph, we perform an ablation study to compare them using the same prompt selection technologies on the provided template pool. In Figure 6, we report the worst-class accuracy with different numbers of enriched categories on ImageNet. From the results, we can find that when enriching a small proportion of the worst-performing categories (less than 30% of all categories), the worst@$k$ accuracy has an obvious increase, demonstrating the

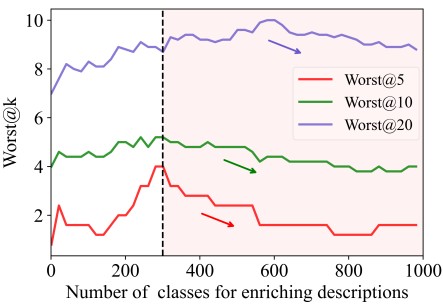

Figure 6: Worst-class performance with different numbers of enriched categories.

effectiveness of enriching descriptions for underperforming categories. When there are more categories enriched, the worst@$k$ would not be further improved but even reduced. This is because when the number of enriched categories increases, more of them are from the well-performing ones, while the underperforming categories are not specially strengthened, and may even be squeezed down. In all of the implementations of CPE, we default to enrich descriptions for the categories with the worst-10% pseudo CMM.

## 7 Conclusion

In this work, we call attention to the worst-performing categories of CLIP models, which are ignored in the context of continuously improving overall performance. To address this issue, we present the CMM and analyze its effectiveness for identifying the worst-performing categories and estimating the potential performance of candidate prompts. Based on CMM, we propose a simple method, CPE, by leveraging large language models to enhance descriptions of worst categories and develop a weighted ensemble to emphasize efficient prompts. The effectiveness of CPE is clearly verified in the experiments across a series of benchmarks. We believe that CPE may provide a promising way to extend the applications of CLIP and make vision-language models more practical.

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

## A CPE FRAMEWORK

CPE has been well presented in the main paper. Here, we further provide the overall framework:

---
**Algorithm 1** The CPE Framework

---
**Input:** Test data set $D$ and a template pool $P$.
    **for** $\text{template}_i$ in $P$ **do**
        Calculate the pseudo label of each sample in $D$.
        Calculate CMM following Equation 6 and locate categories with worst-$k$ CMM.
        Description augmentation for these categories.
        Re-calculate CMM and calculate $w_i$ following Equation 9.
    **end for**
    Get the prediction of each sample by Equation 10.
**Output:** Zero-shot predictions

---

## B IMPLEMENTATION DETAILS

We follow the raw paper of CLIP to implement our methods. The public pre-trained CLIP model is available at `https://github.com/openai/CLIP`. We reproduce all the baselines via Pytorch. The recently proposed ZPE (Allingham et al., 2023) needs 20k images from the pre-training dataset, LAION400m. Unfortunately, the LAION400m dataset requires downloading through a web crawler, which may lead to some inconsistencies due to network information updates. We followed the instructions for using LAION400m and downloaded 1 million images for the ZPE algorithm. Despite our best efforts, the reproduced overall results are still slightly weaker than the results of the original paper. However, it is worth noting that in this work, we focus more on the worst-performing categories. Our proposed CPE shows a significant improvement compared to ZPE and previous methods, which is sufficient to validate the effectiveness of our CPE.

## C THE TEMPLATE POOL

In all implementations of CPE, we utilize the template pool presented by Radford et al. (2021), which can be referred to at `https://github.com/openai/CLIP/blob/main/data/prompts.md` and `https://github.com/openai/CLIP/blob/main/notebooks/Prompt_Engineering_for_ImageNet.ipynb`. For the convenience of a comprehensive overview, we also presents the template pool for each dataset below:

The template pool for ImageNet:

> *a bad photo of a {}. · a photo of many {}. · a sculpture of a {}. · a photo of the hard to see {}. · a low resolution photo of the {}. · a rendering of a {}. · graffiti of a {}. · a bad photo of the {}. · a cropped photo of the {}. · a tattoo of a {}. · the embroidered {}. · a photo of a hard to see {}. · a bright photo of a {}. · a photo of a clean {}. · a photo of a dirty {}. · a dark photo of the {}. · a drawing of a {}. · a photo of my {}. · the plastic {}. · a photo of the cool {}. · a close-up photo of a {}. · a black and white photo of the {}. · a painting of the {}. · a painting of a {}. · a pixelated photo of the {}. · a sculpture of the {}. · a bright photo of the {}. · a cropped photo of a {}. · a plastic {}. · a photo of the dirty {}. · a jpeg corrupted photo of a {}. · a blurry photo of the {}. · a photo of the {}. · a good photo of the {}. · a rendering of the {}. · a {} in a video game. · a photo of one {}. · a doodle of a {}. · a close-up photo of the {}. · a photo of a {}. · the origami {}. · the {} in a video game. · a sketch of a {}. · a doodle of the {}. · a origami {}. · a low resolution photo of a {}. · the toy {}. · a rendition of the {}. · a photo of the clean {}. · a photo of a large {}. · a rendition of a {}. · a photo of a nice {}. · a photo of a weird {}. · a blurry photo of a {}. · a cartoon {}. · art of a {}. · a sketch of the {}. · a embroidered {}. · a pixelated photo of a {}. · itap of the {}. · a jpeg corrupted photo of the {}. · a good photo of a {}. · a plushie {}. · a photo of the nice {}. · a photo of the small {}. · a photo of the weird {}. · the cartoon {}. · art of the {}. · a drawing of the {}. · a photo of the large {}. · a black and white photo of a {}. · the plushie {}. · a dark photo of a {}. · itap of a {}. · graffiti of the {}. · a toy {}. · itap of my {}. · a photo of a cool {}. · a photo of a small {}. · a tattoo of the {}.*

The template pool for CIFAR100:

> *a photo of a {}. · a blurry photo of a {}. · a black and white photo of a {}. · a low contrast photo of a {}. · a high contrast photo of a {}. · a bad photo of a {}. · a good photo of a {}. · a photo of a small {}. · a photo of a big {}. · a photo of the {}. · a blurry photo of the {}. · a black and white photo of the {}. · a low contrast photo of the {}. · a high contrast photo of the {}. · a bad photo of the {}. · a good photo of the {}. · a photo of the small {}. · a photo of the big {}.*

The template pool for Caltech101:

> *a photo of a {}. · a painting of a {}. · a plastic {}. · a sculpture of a {}. · a sketch of a {}. · a tattoo of a {}. · a toy {}. · a rendition of a {}. · a embroidered {}. · a cartoon {}. · a {} in a video game. · a plushie {}. · a origami {}. · art of a {}. · graffiti of a {}. · a drawing of a {}. · a doodle of a {}. · a photo of the {}. · a painting of the {}. · the plastic {}. · a sculpture of the {}. · a sketch of the {}. · a tattoo of the {}. · the toy {}. · a rendition of the {}. · the embroidered {}. · the cartoon {}. · the {} in a video game. · the plushie {}. · the origami {}. · art of the {}. · graffiti of the {}. · a drawing of the {}. · a doodle of the {}.*

The template pool for Cars:

> *a photo of a {}. · a photo of the {}. · a photo of my {}. · i love my {}! · a photo of my dirty {}. · a photo of my clean {}. · a photo of my new {}. · a photo of my old {}.*

The template pool for DTD:

> *a photo of a {} texture. · a photo of a {} pattern. · a photo of a {} thing. · a photo of a {} object. · a photo of the {} texture. · a photo of the {} pattern. · a photo of the {} thing. · a photo of the {} object.*

The template pool for Eurosat:

> *a centered satellite photo of {}. · a centered satellite photo of a {}. · a centered satellite photo of the {}.*

The template pool for Food:

> *a photo of {}, a type of food. · a photo of a {}, a type of food.*

The template pool for Flowers:

> *a photo of {}, a type of flower. · a photo of a {}, a type of flower.*

The template pool for Pets:

> *a photo of {}, a type of pet. · a photo of a {}, a type of pet.*

The template pool for Resisc45:

> *satellite imagery of {}. · aerial imagery of {}. · satellite photo of {}. · aerial photo of {}. · satellite view of {}. · aerial view of {}. · satellite imagery of a {}. · aerial imagery of a {}. · satellite photo of a {}. · aerial photo of a {}. · satellite view of a {}. · aerial view of a {}. · satellite imagery of the {}. · aerial imagery of the {}. · satellite photo of the {}. · aerial photo of the {}. · satellite view of the {}. · aerial view of the {}.*

The template pool for SUN397:

> *a photo of a {}. · a photo of the {}.*

The template pool for FGVCAircraft:

*a photo of a {}, a type of aircraft. · a photo of the {}, a type of aircraft.*

## D  PERFORMANCE OF DIFFERENT TEXTUAL TEMPLATES

In section 3.2, we demonstrate the performances of different textual templates on 8 datasets. Here we provide all the results on all the datasets in Figure 7.

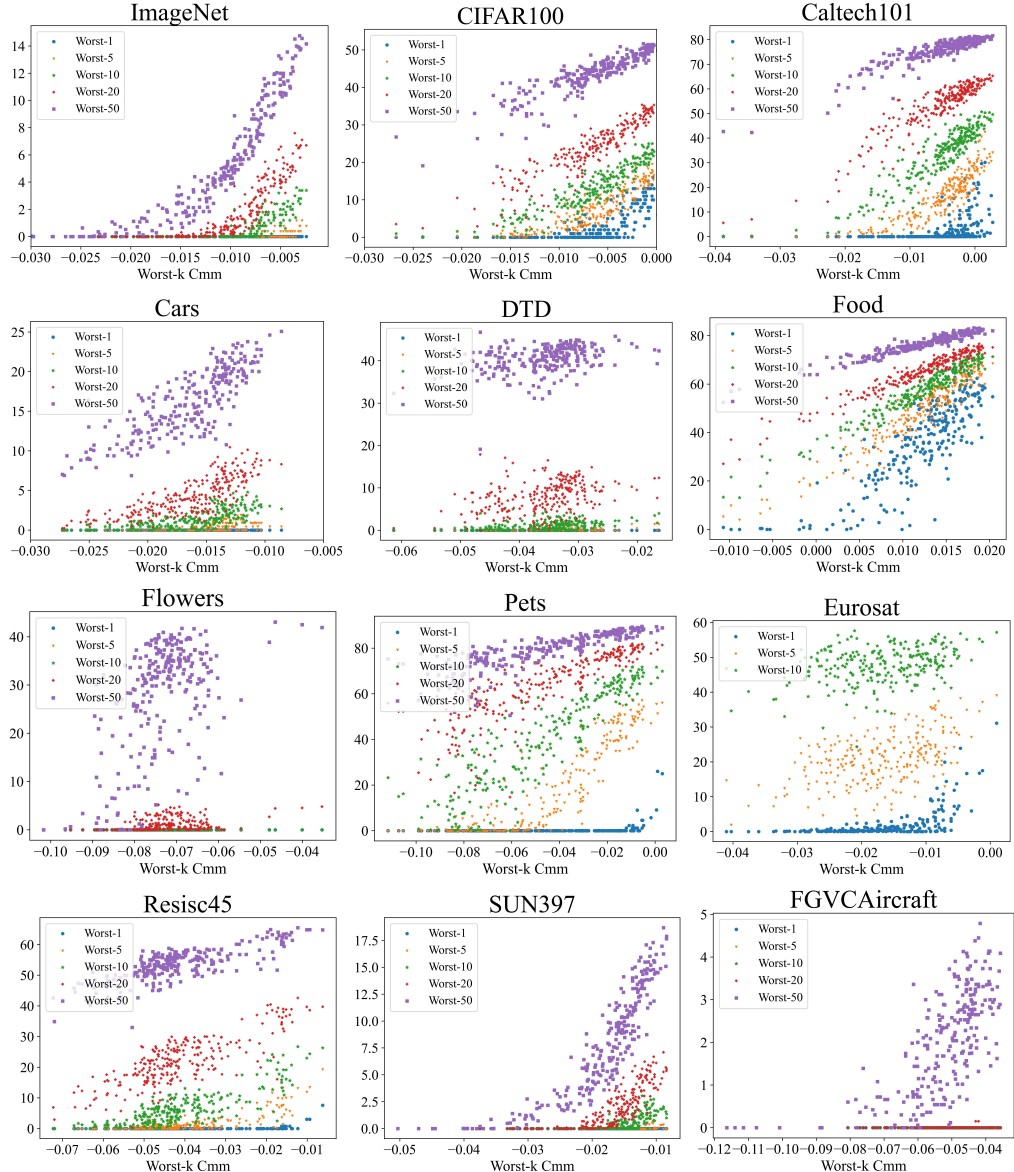

Figure 7: Performance of different textual templates on all 12 datasets with the ViT-B/16 backbone. Each scatter represents a specific textual template. The value of Worst-$k$ CMM ($k = 0.1N$) is approximately in direct proportion to the worst-category performances.

## E  ADDITION EXPERIMENTS

In this section, we report detailed results on 11 fine-grained and specialized datasets: Caltech101 (Fei-Fei et al., 2004), Cars196 (Krause et al., 2013), CIFAR100 (Krizhevsky et al., 2009), DTD (Cimpoi

et al., 2014), EuroSat (Helber et al., 2019), FGVCAircraft (Maji et al., 2013), Food-101 (Kaur et al., 2017), Oxford-Flowers (Nilsback & Zisserman, 2008), Oxford-Pets (Parkhi et al., 2012), Resisc45 (Cheng et al., 2017), and SUN397 (Xiao et al., 2010). Results are shown from Table 4 to 14.

Table 4: Zero-shot Performance (%) on CIFAR100.

| Methods | CLIP ViT-B/16 | | | | | | | |
| | Worst@1 | Worst@5 | Worst@10 | Worst@20 | Worst@50 | HM | GM | Overall |
|---|---|---|---|---|---|---|---|---|
| Classname | 3.00 | 7.00 | 13.60 | 24.30 | 42.48 | 37.67 | 53.72 | 61.57 |
| 'A photo of a {}.' | 13.00 | 16.60 | 23.30 | 35.60 | 52.88 | 56.49 | 63.80 | 68.39 |
| Descriptor | 11.00 | 13.80 | 20.10 | 33.25 | 51.64 | 53.34 | 62.19 | 67.34 |
| Prompt Ens. | 10.00 | 19.60 | 25.90 | 36.70 | 52.46 | 57.73 | 64.37 | 68.61 |
| MLS | 10.00 | 19.40 | 25.50 | 36.55 | 52.44 | 57.63 | 64.34 | 68.61 |
| ZPE* | 10.00 | 19.20 | 25.40 | 36.80 | 52.52 | 57.60 | 64.37 | **68.67** |
| CPE† | 10.00 | 19.20 | 25.40 | 36.70 | 52.46 | 57.55 | 64.32 | 68.62 |
| CPE‡ | **17.00** | 21.00 | 29.00 | **38.00** | **53.20** | **59.77** | **64.93** | 68.63 |
| CPE | **17.00** | **22.20** | **29.20** | 37.45 | 52.86 | 59.72 | 64.81 | 68.52 |
| Methods | CLIP ViT-L/14 | | | | | | | |
| | Worst@1 | Worst@5 | Worst@10 | Worst@20 | Worst@50 | HM | GM | Overall |
| Classname | 70.60 | 85.78 | 91.40 | 91.40 | 91.40 | 90.57 | 91.01 | 91.40 |
| 'A photo of a {}.' | 85.90 | 93.12 | 95.15 | 95.15 | 95.15 | 95.02 | 95.09 | 95.15 |
| Descriptor | **92.00** | 92.96 | 94.83 | 94.83 | 94.83 | 94.78 | 94.81 | 94.83 |
| Prompt Ens. | 90.50 | **94.16** | 95.60 | 95.60 | 95.60 | 95.55 | 95.57 | 95.60 |
| MLS | 90.40 | 94.14 | 95.58 | 95.58 | 95.58 | 95.53 | 95.55 | 95.58 |
| ZPE* | 90.30 | 94.08 | 95.56 | 95.56 | 95.56 | 95.51 | 95.53 | 95.56 |
| CPE† | 90.60 | 94.14 | **95.77** | **95.77** | **95.77** | **95.71** | **95.74** | **95.77** |
| CPE‡ | 90.40 | 93.92 | 95.53 | 95.53 | 95.53 | 95.47 | 95.50 | 95.53 |
| CPE | 88.30 | 93.38 | 95.31 | 95.31 | 95.31 | 95.23 | 95.27 | 95.31 |

Table 5: Zero-shot Performance (%) on Caltech101.

| Methods | CLIP ViT-B/16 | | | | | | | |
| | Worst@1 | Worst@5 | Worst@10 | Worst@20 | Worst@50 | HM | GM | Overall |
|---|---|---|---|---|---|---|---|---|
| Classname | 0.00 | 24.06 | **43.21** | 60.81 | 78.88 | 0.00 | 0.00 | 85.06 |
| 'A photo of a {}.' | 0.00 | 17.43 | 37.79 | 57.89 | 78.60 | 0.00 | 0.00 | 87.65 |
| Descriptor | 7.59 | 20.38 | 41.41 | **61.32** | 80.32 | 74.92 | 85.81 | **88.88** |
| Prompt Ens. | 17.24 | 22.85 | 39.87 | 61.04 | **80.60** | 78.80 | 86.34 | 87.68 |
| MLS | 17.54 | 23.04 | 40.10 | 61.05 | 80.57 | 79.02 | 86.37 | 87.69 |
| ZPE* | 16.09 | 23.11 | 39.93 | 60.87 | 80.55 | 78.74 | 86.33 | 87.68 |
| CPE† | **20.18** | **24.07** | 40.06 | 61.03 | 80.58 | **79.83** | **86.54** | 87.70 |
| CPE‡ | 5.71 | 19.57 | 40.39 | 59.31 | 79.04 | 71.24 | 84.60 | 87.68 |
| CPE | 8.57 | 22.26 | 41.64 | 60.56 | 79.31 | 74.30 | 85.10 | 87.66 |
| Methods | CLIP ViT-L/14 | | | | | | | |
| | Worst@1 | Worst@5 | Worst@10 | Worst@20 | Worst@50 | HM | GM | Overall |
| Classname | 5.29 | 23.36 | 43.28 | 63.61 | 82.44 | 68.58 | 86.36 | 84.56 |
| 'A photo of a {}.' | 0.00 | 45.66 | 61.59 | 74.74 | 87.53 | 0.00 | 0.00 | 91.13 |
| Descriptor | **33.33** | 41.43 | 55.42 | 71.50 | 86.34 | 89.18 | 91.64 | **91.85** |
| Prompt Ens. | 10.34 | 45.31 | 59.07 | 74.72 | **87.80** | 86.12 | 91.96 | 90.58 |
| MLS | 9.66 | 45.45 | 59.35 | 74.63 | 87.79 | 85.64 | 91.91 | 90.58 |
| ZPE* | 10.80 | 45.18 | 59.06 | 74.67 | 87.77 | 86.42 | **91.99** | 90.56 |
| CPE† | 8.28 | 44.87 | 59.32 | 74.57 | 87.74 | 84.37 | 91.75 | 90.55 |
| CPE‡ | 9.43 | **48.90** | **63.36** | **75.66** | 87.67 | **85.71** | 91.94 | 91.78 |
| CPE | 2.30 | 50.22 | 63.04 | 75.27 | 87.67 | 67.16 | 90.82 | 91.48 |

Table 6: Zero-shot Performance (%) on Cars.

| Methods | CLIP ViT-B/16 | | | | | | | |
| | Worst@1 | Worst@5 | Worst@10 | Worst@20 | Worst@50 | HM | GM | Overall |
|---|---|---|---|---|---|---|---|---|
| Classname | 0.00 | 0.00 | 0.94 | 3.53 | 16.68 | 0.00 | 0.00 | 61.53 |
| 'A photo of a {}.' | 0.00 | 1.48 | 3.23 | 7.46 | 20.61 | 0.00 | 0.00 | 63.51 |
| Descriptor | 0.00 | 0.91 | 2.92 | 6.73 | 18.29 | 0.00 | 0.00 | 63.25 |
| Prompt Ens. | 0.00 | 0.47 | 2.60 | 8.47 | 23.92 | 0.00 | 0.00 | 64.54 |
| MLS | 0.00 | 0.47 | 2.36 | 8.47 | **24.11** | 0.00 | 0.00 | 64.52 |
| ZPE* | 0.00 | 0.47 | 2.36 | 8.25 | 23.84 | 0.00 | 0.00 | 64.46 |
| CPE† | 0.00 | 0.93 | 2.72 | 8.58 | 24.16 | 0.00 | 0.00 | **64.67** |
| CPE‡ | 0.00 | **2.10** | **4.16** | **8.76** | 20.87 | 0.00 | 0.00 | 63.71 |
| CPE | 0.00 | 1.45 | 3.65 | 7.51 | 21.02 | 0.00 | 0.00 | 63.15 |
| Methods | CLIP ViT-L/14 | | | | | | | |
| | Worst@1 | Worst@5 | Worst@10 | Worst@20 | Worst@50 | HM | GM | Overall |
| Classname | 0.00 | 3.33 | 5.26 | 11.48 | 32.90 | 0.00 | 0.00 | 74.48 |
| 'A photo of a {}.' | 0.00 | 2.78 | 9.06 | 19.88 | 38.80 | 0.00 | 0.00 | 76.41 |
| Descriptor | 0.00 | 2.41 | 5.33 | 12.75 | 35.70 | 0.00 | 0.00 | 75.49 |
| Prompt Ens. | **2.78** | **10.42** | **15.81** | 24.06 | **44.06** | **56.43** | **71.40** | **77.66** |
| MLS | **2.78** | **10.42** | 15.35 | 23.83 | 44.03 | 56.31 | 71.37 | 77.66 |
| ZPE* | **2.78** | **10.42** | 15.58 | **24.07** | 44.05 | 56.39 | 71.37 | 77.64 |
| CPE† | **2.78** | **10.42** | 15.35 | 23.72 | 43.93 | 56.24 | 71.32 | 77.64 |
| CPE‡ | 2.50 | 6.17 | 13.01 | 22.48 | 40.23 | 48.76 | 68.99 | 76.48 |
| CPE | 2.50 | 7.26 | 14.85 | 23.79 | 40.68 | 50.10 | 69.15 | 76.25 |

Table 7: Zero-shot Performance (%) on DTD.

| Methods | CLIP ViT-B/16 | | | | | | | |
| | Worst@1 | Worst@5 | Worst@10 | Worst@20 | Worst@50 | HM | GM | Overall |
|---|---|---|---|---|---|---|---|---|
| Classname | 0.00 | 0.00 | 3.50 | 14.12 | 44.57 | 0.00 | 0.00 | 44.57 |
| 'A photo of a {}.' | 0.00 | 0.50 | 2.00 | 12.25 | 42.82 | 0.00 | 0.00 | 42.82 |
| Descriptor | 0.00 | 0.00 | 2.00 | 11.88 | 45.48 | 0.00 | 0.00 | 45.48 |
| Prompt Ens. | 0.00 | 0.50 | 2.00 | 11.25 | 45.64 | 0.00 | 0.00 | 45.64 |
| MLS | 0.00 | 0.50 | 2.00 | 11.38 | 45.69 | 0.00 | 0.00 | 45.69 |
| ZPE* | 0.00 | 1.00 | 2.25 | 11.62 | 45.59 | 0.00 | 0.00 | 45.59 |
| CPE† | 0.00 | 1.00 | 2.25 | 12.12 | 46.22 | 0.00 | 0.00 | 46.22 |
| CPE‡ | 0.00 | 1.00 | 4.25 | **16.50** | **47.13** | 0.00 | 0.00 | **47.13** |
| CPE | 0.00 | **2.00** | **5.75** | 16.38 | 46.86 | 0.00 | 0.00 | 46.86 |
| Methods | CLIP ViT-L/14 | | | | | | | |
| | Worst@1 | Worst@5 | Worst@10 | Worst@20 | Worst@50 | HM | GM | Overall |
| Classname | 0.00 | 1.50 | 6.25 | 19.88 | 50.37 | 0.00 | 0.00 | 50.37 |
| 'A photo of a {}.' | 0.00 | 2.50 | 10.50 | 23.75 | 52.39 | 0.00 | 0.00 | 52.39 |
| Descriptor | 0.00 | 2.50 | 7.75 | 24.00 | 54.15 | 0.00 | 0.00 | 54.15 |
| Prompt Ens. | 0.00 | 4.50 | 10.00 | 23.62 | 55.11 | 0.00 | 0.00 | 55.11 |
| MLS | 0.00 | 4.50 | 10.00 | 23.62 | 55.11 | 0.00 | 0.00 | 55.11 |
| ZPE* | 0.00 | 4.50 | 9.50 | 23.62 | 55.11 | 0.00 | 0.00 | 55.11 |
| CPE† | 0.00 | 4.50 | 9.50 | 23.75 | 55.32 | 0.00 | 0.00 | 55.32 |
| CPE‡ | 0.00 | **6.00** | **12.25** | **26.38** | **55.64** | 0.00 | 0.00 | **55.64** |
| CPE | 0.00 | 5.50 | 11.75 | 25.38 | 55.27 | 0.00 | 0.00 | 55.27 |

Table 8: Zero-shot Performance (%) on Eurosat.

| Methods | CLIP ViT-B/16 | | | | | | | |
| | Worst@1 | Worst@5 | Worst@10 | Worst@20 | Worst@50 | HM | GM | Overall |
|---|---|---|---|---|---|---|---|---|
| Classname | 3.00 | 14.93 | 43.00 | 43.00 | 43.00 | 11.39 | 25.19 | 44.18 |
| 'A photo of a {}.' | 0.03 | 18.93 | 48.27 | 48.27 | 48.27 | 0.31 | 14.79 | 50.15 |
| Descriptor | 0.50 | 27.20 | 48.01 | 48.01 | 48.01 | 4.51 | 30.19 | 49.06 |
| Prompt Ens. | 13.96 | 26.69 | 53.39 | 53.39 | 53.39 | 33.85 | 43.44 | 53.55 |
| MLS | 13.52 | 26.55 | 53.32 | 53.32 | 53.32 | 33.55 | 43.27 | 53.48 |
| ZPE* | 14.08 | 26.68 | 53.40 | 53.40 | 53.40 | 33.88 | 43.46 | 53.55 |
| CPE† | 16.72 | 28.53 | 54.27 | 54.27 | 54.27 | 36.88 | 45.47 | 54.44 |
| CPE‡ | 17.10 | 27.66 | 54.00 | 54.00 | 54.00 | 38.23 | 45.89 | 53.77 |
| CPE | **18.40** | **30.16** | **55.03** | **55.03** | **55.03** | **40.58** | **47.68** | **54.72** |
| Methods | CLIP ViT-L/14 | | | | | | | |
| | Worst@1 | Worst@5 | Worst@10 | Worst@20 | Worst@50 | HM | GM | Overall |
| Classname | 4.13 | 19.13 | 41.69 | 41.69 | 41.69 | 18.80 | 30.56 | 39.51 |
| 'A photo of a {}.' | 0.00 | 29.80 | 57.23 | 57.23 | 57.23 | 0.00 | 0.00 | 55.95 |
| Descriptor | 3.47 | 30.74 | 55.09 | 55.09 | 55.09 | 21.21 | 41.90 | 53.36 |
| Prompt Ens. | 8.87 | 37.82 | 61.39 | 61.39 | 61.39 | 34.32 | 49.80 | 60.07 |
| MLS | 8.80 | 37.82 | 61.38 | 61.38 | 61.38 | 34.23 | 49.77 | 60.06 |
| ZPE* | 8.80 | 37.89 | 61.43 | 61.43 | 61.43 | 34.39 | 49.89 | 60.11 |
| CPE† | 10.60 | 38.75 | 61.85 | 61.85 | 61.85 | 37.28 | 51.22 | 60.58 |
| CPE‡ | **16.37** | **42.79** | **63.63** | **63.63** | **63.63** | **48.15** | **56.84** | **63.63** |
| CPE | 12.90 | 38.79 | 63.07 | 63.07 | 63.07 | 41.81 | 53.65 | 62.99 |

Table 9: Zero-shot Performance (%) on Food.

| Methods | CLIP ViT-B/16 | | | | | | | |
| | Worst@1 | Worst@5 | Worst@10 | Worst@20 | Worst@50 | HM | GM | Overall |
|---|---|---|---|---|---|---|---|---|
| Classname | 20.80 | 51.76 | 62.20 | 70.18 | 79.70 | 83.80 | 85.80 | 86.89 |
| 'A photo of a {}.' | 51.20 | 63.60 | 68.72 | 74.18 | 82.03 | 87.35 | 87.92 | 88.42 |
| Descriptor | **58.40** | 69.36 | 72.88 | 76.50 | 82.64 | 88.05 | 88.45 | 88.81 |
| Prompt Ens. | 37.60 | 64.72 | 71.04 | 75.90 | 82.81 | 87.71 | 88.45 | 88.99 |
| MLS | 37.60 | 64.80 | 71.12 | 75.98 | 82.83 | 87.73 | 88.47 | 89.01 |
| ZPE* | 37.60 | 64.72 | 71.04 | 75.90 | 82.81 | 87.72 | 88.46 | 89.00 |
| CPE† | 37.60 | 64.88 | 71.20 | 76.04 | 82.84 | 87.74 | 88.47 | 89.01 |
| CPE‡ | 54.40 | **69.84** | **73.44** | **77.00** | **83.27** | **88.43** | **88.84** | **89.21** |
| CPE | 54.40 | **69.84** | **73.44** | **77.00** | **83.27** | **88.43** | **88.84** | **89.21** |
| Methods | CLIP ViT-L/14 | | | | | | | |
| | Worst@1 | Worst@5 | Worst@10 | Worst@20 | Worst@50 | HM | GM | Overall |
| Classname | 62.40 | 71.04 | 76.00 | 81.08 | 87.40 | 91.55 | 91.87 | 92.16 |
| 'A photo of a {}.' | 58.40 | 74.32 | 78.48 | 82.66 | 88.18 | 92.03 | 92.31 | 92.55 |
| Descriptor | 46.40 | 68.88 | 76.12 | 82.04 | 88.36 | 91.93 | 92.42 | 92.80 |
| Prompt Ens. | 62.00 | 75.12 | 80.92 | 84.80 | 89.38 | 92.86 | 93.11 | 93.33 |
| MLS | 62.00 | 75.12 | 80.92 | 84.80 | 89.38 | 92.87 | 93.12 | 93.33 |
| ZPE* | 62.40 | 75.12 | **80.96** | 84.82 | 89.38 | 92.88 | 93.12 | 93.33 |
| CPE† | 62.40 | 75.20 | **80.96** | 84.82 | 89.36 | 92.87 | 93.11 | 93.32 |
| CPE‡ | **73.20** | **75.76** | **80.96** | **84.94** | **89.49** | **93.01** | **93.19** | **93.36** |
| CPE | **73.20** | **75.76** | **80.96** | **84.94** | **89.49** | **93.01** | **93.19** | **93.36** |

Table 10: Zero-shot Performance (%) on Flowers.

| Methods | CLIP ViT-B/16 | | | | | | | |
| | Worst@1 | Worst@5 | Worst@10 | Worst@20 | Worst@50 | HM | GM | Overall |
|---|---|---|---|---|---|---|---|---|
| Classname | 0.00 | 0.00 | 0.00 | 0.39 | 35.36 | 0.00 | 0.00 | 63.64 |
| 'A photo of a {}.' | 0.00 | 0.00 | 0.00 | 1.98 | 38.28 | 0.00 | 0.00 | 67.15 |
| Descriptor | 0.00 | 0.00 | 0.00 | 2.02 | 41.58 | 0.00 | 0.00 | 70.91 |
| Prompt Ens. | 0.00 | 0.00 | 0.00 | 5.29 | 44.29 | 0.00 | 0.00 | 72.14 |
| MLS | 0.00 | 0.00 | 0.00 | **5.46** | **44.38** | 0.00 | 0.00 | **72.16** |
| ZPE* | 0.00 | 0.00 | 0.00 | 5.07 | 44.17 | 0.00 | 0.00 | 72.08 |
| CPE† | 0.00 | 0.00 | 0.00 | 5.08 | 44.30 | 0.00 | 0.00 | 71.98 |
| CPE‡ | 0.00 | 0.00 | 0.00 | 4.40 | 43.36 | 0.00 | 0.00 | 71.57 |
| CPE | 0.00 | 0.00 | 0.00 | 4.40 | 43.36 | 0.00 | 0.00 | 71.57 |
| Methods | CLIP ViT-L/14 | | | | | | | |
| | Worst@1 | Worst@5 | Worst@10 | Worst@20 | Worst@50 | HM | GM | Overall |
| Classname | 0.00 | 0.00 | 0.00 | 10.00 | 51.15 | 0.00 | 0.00 | 71.87 |
| 'A photo of a {}.' | 0.00 | 0.00 | 0.00 | 11.66 | 52.80 | 0.00 | 0.00 | 74.26 |
| Descriptor | 0.00 | 0.00 | 0.00 | 10.30 | 54.84 | 0.00 | 0.00 | 77.77 |
| Prompt Ens. | 0.00 | **0.85** | **4.14** | 20.83 | 58.79 | 0.00 | 0.00 | 78.92 |
| MLS | 0.00 | **0.85** | **4.14** | **20.97** | **59.00** | 0.00 | 0.00 | **78.96** |
| ZPE* | 0.00 | **0.85** | **4.14** | 20.90 | 58.92 | 0.00 | 0.00 | 78.94 |
| CPE† | 0.00 | **0.85** | **4.14** | 20.77 | 58.97 | 0.00 | 0.00 | 78.94 |
| CPE‡ | 0.00 | 0.00 | 1.65 | 18.81 | 57.61 | 0.00 | 0.00 | 78.44 |
| CPE | 0.00 | 0.00 | 1.65 | 18.81 | 57.61 | 0.00 | 0.00 | 78.44 |

Table 11: Zero-shot Performance (%) on Pets.

| Methods | CLIP ViT-B/16 | | | | | | | |
| | Worst@1 | Worst@5 | Worst@10 | Worst@20 | Worst@50 | HM | GM | Overall |
|---|---|---|---|---|---|---|---|---|
| Classname | 0.00 | 14.80 | 44.66 | 67.75 | 81.62 | 0.00 | 0.00 | 81.85 |
| 'A photo of a {}.' | 0.00 | 50.00 | 66.00 | 79.29 | 87.80 | 0.00 | 0.00 | 88.06 |
| Descriptor | 6.82 | 44.76 | 62.43 | 77.25 | 86.67 | 65.01 | 81.53 | 86.92 |
| Prompt Ens. | 31.82 | 53.56 | 68.80 | 80.59 | 88.45 | 82.43 | 86.16 | 88.63 |
| MLS | 31.82 | 53.76 | 68.90 | 80.59 | 88.45 | 82.58 | 86.20 | 88.63 |
| ZPE* | 31.82 | 53.76 | 68.90 | 80.64 | 88.48 | 82.60 | 86.23 | 88.66 |
| CPE† | 31.82 | 53.76 | 68.80 | 80.75 | 88.54 | 82.65 | 86.28 | 88.72 |
| CPE‡ | **38.64** | **59.13** | **70.98** | **81.80** | **89.08** | **85.68** | **87.64** | **89.23** |
| CPE | **38.64** | **59.13** | **70.98** | **81.80** | **89.08** | **85.68** | **87.64** | **89.23** |
| Methods | CLIP ViT-L/14 | | | | | | | |
| | Worst@1 | Worst@5 | Worst@10 | Worst@20 | Worst@50 | HM | GM | Overall |
| Classname | 0.00 | 47.20 | 63.71 | 78.39 | 87.94 | 0.00 | 0.00 | 88.23 |
| 'A photo of a {}.' | 57.00 | 72.37 | 80.92 | 88.12 | 93.12 | 91.87 | 92.55 | 93.19 |
| Descriptor | 45.45 | 66.29 | 77.27 | 85.73 | 91.83 | 88.97 | 90.65 | 91.99 |
| Prompt Ens. | **59.00** | **72.69** | 82.00 | **89.15** | **93.81** | 92.60 | **93.26** | **93.81** |
| MLS | **59.00** | **72.69** | 82.10 | **89.15** | **93.81** | **92.61** | **93.26** | **93.81** |
| ZPE* | **59.00** | 72.49 | 81.90 | 89.10 | 93.78 | 92.56 | 93.22 | 93.79 |
| CPE† | **59.00** | 72.49 | 81.90 | 89.05 | 93.75 | 92.54 | 93.20 | 93.76 |
| CPE‡ | 57.00 | 70.49 | 81.13 | 88.66 | 93.52 | 92.09 | 92.87 | 93.51 |
| CPE | 57.00 | 70.49 | 81.13 | 88.66 | 93.52 | 92.09 | 92.87 | 93.51 |

Table 12: Zero-shot Performance (%) on Resisc45.

| Methods | CLIP ViT-B/16 | | | | | | | |
| | Worst@1 | Worst@5 | Worst@10 | Worst@20 | Worst@50 | HM | GM | Overall |
|---|---|---|---|---|---|---|---|---|
| Classname | 0.00 | 2.23 | 7.75 | 26.48 | 54.90 | 0.00 | 0.00 | 54.70 |
| 'A photo of a {}.' | 0.00 | 1.26 | 6.14 | 25.26 | 55.79 | 0.00 | 0.00 | 55.75 |
| Descriptor | 0.00 | 1.78 | 8.60 | 26.99 | 58.18 | 0.00 | 0.00 | 57.95 |
| Prompt Ens. | 0.00 | 12.97 | 24.93 | 40.74 | 65.25 | 0.00 | 0.00 | 65.41 |
| MLS | 0.00 | 13.13 | 24.80 | 40.59 | 65.28 | 0.00 | 0.00 | 65.44 |
| ZPE* | 0.00 | 12.67 | 24.32 | 40.55 | 65.16 | 0.00 | 0.00 | 65.32 |
| CPE† | 0.00 | **13.27** | **25.41** | **40.84** | **65.32** | 0.00 | 0.00 | **65.46** |
| CPE‡ | 0.00 | 5.99 | 22.39 | 40.40 | 64.39 | 0.00 | 0.00 | 64.63 |
| CPE | 0.00 | 7.09 | 21.66 | 40.29 | 64.73 | 0.00 | 0.00 | 64.94 |
| Methods | CLIP ViT-L/14 | | | | | | | |
| | Worst@1 | Worst@5 | Worst@10 | Worst@20 | Worst@50 | HM | GM | Overall |
| Classname | 0.00 | 1.86 | 8.96 | 30.35 | 59.79 | 0.00 | 0.00 | 59.41 |
| 'A photo of a {}.' | 0.00 | 4.44 | 16.10 | 34.63 | 63.36 | 0.00 | 0.00 | 63.37 |
| Descriptor | 0.00 | 4.43 | 15.01 | 32.80 | 63.13 | 0.00 | 0.00 | 62.86 |
| Prompt Ens. | 0.79 | 16.73 | 29.93 | 48.21 | 70.59 | 22.26 | 60.39 | 70.67 |
| MLS | 0.79 | **17.01** | 30.01 | 48.18 | 70.55 | 22.30 | **60.43** | 70.63 |
| ZPE* | 0.79 | 16.88 | 29.88 | 48.15 | 70.54 | 22.27 | 60.38 | 70.62 |
| CPE† | 0.79 | 16.00 | 30.01 | 48.22 | 70.57 | 22.18 | 60.26 | 70.67 |
| CPE‡ | 0.79 | 12.06 | 29.64 | 48.20 | 70.37 | 17.15 | 57.21 | 70.65 |
| CPE | **1.54** | 13.47 | **31.37** | **49.61** | **70.99** | **22.50** | 58.63 | **71.32** |

Table 13: Zero-shot Performance (%) on SUN397.

| Methods | CLIP ViT-B/16 | | | | | | | |
| | Worst@1 | Worst@5 | Worst@10 | Worst@20 | Worst@50 | HM | GM | Overall |
|---|---|---|---|---|---|---|---|---|
| Classname | 0.00 | 0.00 | 0.17 | 1.82 | 10.57 | 0.00 | 0.00 | 60.65 |
| 'A photo of a {}.' | 0.00 | 0.00 | 1.08 | 4.44 | 14.36 | 0.00 | 0.00 | 60.34 |
| Descriptor | 0.00 | 0.00 | 0.78 | 3.48 | 13.80 | 0.00 | 0.00 | 64.06 |
| Prompt Ens. | 0.00 | 0.00 | 1.40 | **6.32** | 18.71 | 0.00 | 0.00 | 63.75 |
| MLS | 0.00 | 0.00 | 1.40 | 6.25 | **18.72** | 0.00 | 0.00 | 63.75 |
| ZPE* | 0.00 | 0.00 | 1.40 | 6.22 | 18.86 | 0.00 | 0.00 | 63.75 |
| CPE† | 0.00 | 0.00 | 1.40 | 6.25 | 18.76 | 0.00 | 0.00 | 63.75 |
| CPE‡ | 0.00 | 0.00 | **1.44** | 5.83 | 18.08 | 0.00 | 0.00 | **64.23** |
| CPE | 0.00 | 0.00 | **1.44** | 5.83 | 18.08 | 0.00 | 0.00 | **64.23** |
| Methods | CLIP ViT-L/14 | | | | | | | |
| | Worst@1 | Worst@5 | Worst@10 | Worst@20 | Worst@50 | HM | GM | Overall |
| Classname | 0.00 | 0.00 | 1.07 | 2.85 | 11.99 | 0.00 | 0.00 | 63.72 |
| 'A photo of a {}.' | 0.00 | 0.00 | 1.58 | 5.32 | 16.41 | 0.00 | 0.00 | 64.42 |
| Descriptor | 0.00 | 0.00 | 2.33 | 6.64 | 17.66 | 0.00 | 0.00 | 67.51 |
| Prompt Ens. | 0.00 | 0.00 | **2.28** | 7.98 | 19.10 | 0.00 | 0.00 | 67.32 |
| MLS | 0.00 | 0.00 | **2.28** | 7.95 | 19.04 | 0.00 | 0.00 | 67.31 |
| ZPE* | 0.00 | 0.00 | **2.28** | 7.98 | 19.09 | 0.00 | 0.00 | 67.32 |
| CPE† | 0.00 | 0.00 | **2.28** | **8.05** | 19.25 | 0.00 | 0.00 | 67.38 |
| CPE‡ | 0.00 | 0.00 | 0.81 | 5.47 | **19.93** | 0.00 | 0.00 | 67.60 |
| CPE | 0.00 | 0.00 | 0.81 | 5.47 | **19.93** | 0.00 | 0.00 | **67.60** |

Table 14: Zero-shot Performance (%) on FGVCAircraft.

| Methods | CLIP ViT-B/16 | | | | | | | |
| | Worst@1 | Worst@5 | Worst@10 | Worst@20 | Worst@50 | HM | GM | Overall |
| --- | --- | --- | --- | --- | --- | --- | --- | --- |
| Classname | 0.00 | 0.00 | 0.00 | 0.00 | 2.58 | 0.00 | 0.00 | 21.24 |
| 'A photo of a {}.' | 0.00 | 0.00 | 0.00 | **0.15** | **4.45** | 0.00 | 0.00 | 23.94 |
| Descriptor | 0.00 | 0.00 | 0.00 | 0.00 | 3.01 | 0.00 | 0.00 | 23.40 |
| Prompt Ens. | 0.00 | 0.00 | 0.00 | 0.00 | 3.74 | 0.00 | 0.00 | 24.18 |
| MLS | 0.00 | 0.00 | 0.00 | 0.00 | 3.68 | 0.00 | 0.00 | 24.12 |
| ZPE* | 0.00 | 0.00 | 0.00 | 0.00 | 3.92 | 0.00 | 0.00 | **24.51** |
| CPE† | 0.00 | 0.00 | 0.00 | 0.00 | 4.09 | 0.00 | 0.00 | 24.33 |
| CPE‡ | 0.00 | 0.00 | 0.00 | 0.00 | 4.22 | 0.00 | 0.00 | 24.39 |
| CPE | 0.00 | 0.00 | 0.00 | 0.00 | 4.22 | 0.00 | 0.00 | 24.39 |
| Methods | CLIP ViT-L/14 | | | | | | | |
| | Worst@1 | Worst@5 | Worst@10 | Worst@20 | Worst@50 | HM | GM | Overall |
| Classname | 0.00 | 0.00 | 0.00 | 0.59 | 5.40 | 0.00 | 0.00 | 27.18 |
| 'A photo of a {}.' | 0.00 | 0.00 | 0.00 | 0.44 | 7.37 | 0.00 | 0.00 | 30.33 |
| Descriptor | 0.00 | 0.00 | 0.00 | 0.60 | 8.10 | 0.00 | 0.00 | 30.90 |
| Prompt Ens. | 0.00 | 0.00 | 0.00 | 0.44 | 8.92 | 0.00 | 0.00 | 31.56 |
| MLS | 0.00 | 0.00 | 0.00 | 0.44 | 9.06 | 0.00 | 0.00 | 31.68 |
| ZPE* | 0.00 | 0.00 | 0.00 | 0.44 | 9.08 | 0.00 | 0.00 | 31.80 |
| CPE† | 0.00 | 0.00 | 0.00 | 0.74 | 9.01 | 0.00 | 0.00 | 31.56 |
| CPE‡ | 0.00 | 0.00 | 0.00 | **1.94** | **10.19** | 0.00 | 0.00 | **32.49** |
| CPE | 0.00 | 0.00 | 0.00 | **1.94** | **10.19** | 0.00 | 0.00 | **32.49** |

