# OpenReview forum: "Examining the Achilles' Heel of CLIP Models: The Worst-Performing Categories"
_ICLR.cc/2024/Conference — Submitted to ICLR 2024_

### Official Review · Reviewer_PArF · 2023-10-21

**Soundness:** 3 good
**Presentation:** 3 good
**Contribution:** 3 good
**Rating:** 5
**Confidence:** 4

**Summary:**

The paper discusses a very interesting and overlooked topic: the worst-performing class of CLIP. To mitigate this issue, the authors propose CMM  to measure confusion. Also, the authors query LLM to enrich the descriptions of worst-performing categories. Experiments prove the effectiveness of the proposed method.

**Strengths:**

1. The discussion topic is very interesting and novel for me, I like it;
2. Good writing and smooth description;
3. Good discussion and meaningful visualization.

**Weaknesses:**

1. The contribution is not enough. I fully understand the importance of the discussed topic. But could you show more insight and the importance of your paper?
2. I cannot understand the LLM meaning in your paper: “We further query large language models to enrich descriptions of worst-performing categories.” Can we use other description models? Or can we describe the categories by ourselves? Ablation studies needed?
3. The paper is not well self-defined. For example, what PE in the Figure 1 means?
4. The main discussion of the paper is on CLIP? Have you investigated other methods? Such as BLIP. Do their worst prediction same with CLIP? Or could you ensemble different VL models to solve the problem you discussed?

**Questions:**

I think this paper's score should be between 5 and 6. Please provide a detailed rebuttal to convince me and further clarify your contribution.

---

### Official Review · Reviewer_RDxr · 2023-10-23

**Soundness:** 2 fair
**Presentation:** 3 good
**Contribution:** 2 fair
**Rating:** 5
**Confidence:** 4

**Summary:**

This paper focuses on the zero-shot image classification ability of CLIP. It mainly improves the accuracy of the worst categories. The paper proposes a Class-wise Matching Margin (CMM) to measure the underperforming categories. CMM is calculated based on the similarities of images and textual prompts. It can identify the worst-performing categories and estimate the potential performance of the candidate prompts. Furthermore, large language models are requested to enrich descriptions of worst-performing categories and build a weighted ensemble of prompts. CMM boosts the accuracy on the worst-10 categories on ImageNet to 5.2%, without manual prompt engineering, laborious optimization, or access to labeled validation data.

**Strengths:**

1. This paper is easy to follow.
2. Comprehensive experimental results are provided, and illustrations of the results are great.
3. There are complete implementation details of the method.

**Weaknesses:**

1. The main concern is the technical contribution. CLIP uses image-text similarity to perform zero-shot classification, so evaluating the similarities roughly equals evaluating classification accuracy.
2. Pseudo-CMM is not clearly described. How to set the pseudo label for a sample?
3. The CPE algorithm is conducted on the test set. Although the paper claims the annotations are not used, it still seems inappropriate.

**Questions:**

1. What is the pseudo labels for test set?
2. Does each category have its own ensemble of prompts?

---

### Official Review · Reviewer_7ik6 · 2023-10-30

**Soundness:** 1 poor
**Presentation:** 2 fair
**Contribution:** 1 poor
**Rating:** 3
**Confidence:** 5

**Summary:**

This paper investigates the terrible performance of CLIP models in certain categories of ImageNet and highlights potential risks associated with their applications. To address this issue, the paper proposes to enrich their textual prompts. The Class-wise Matching Margin (CMM) is introduced as a measure of inference confusion, quantifying the margin between the similarity of images to the correct textual prompt and the most susceptible one. Leveraging LLMs, the paper enriches the textual prompts for these categories, leading to improved performance according to the conducted experiments.

**Strengths:**

The examination of the worst-performing categories in ImageNet is intriguing.

**Weaknesses:**

The problem definition lacks clarity and requires further elaboration. Specific information regarding the worst-performing classes in ImageNet is missing, hindering readers from determining whether the poor performance stems from inherent ambiguities within these classes or issues with the model itself. Additionally, the paper neglects to compare CLIP models with models learned using other methods, such as supervised learning or visual contrastive learning. This omission makes it challenging to discern whether the identified problem is exclusive to CLIP models or prevalent across all models, significantly impacting the paper's contribution.

The proposed method is relatively straightforward and does not exhibit notable performance improvements. The idea of enhancing performance through enriched textual prompts is intuitive, resulting in limited novel insights provided by this paper. Furthermore, the analysis of the specific effects of richer textual prompts on the model remains superficial.

In conclusion, this article lacks a clear and concise problem definition, and the underlying causes of the identified problem are unclear. Moreover, the proposed method is relatively trivial, yielding only modest performance improvements. Consequently, the contribution of this article falls short of justifying its publication at ICLR.

**Questions:**

No other questions.

---

### Meta-Review · Area_Chair_2rmL · 2023-12-05

**Metareview:**

The paper examines the performance of the CLIP model on image recognition. The authors show that using LLMs to enhance the descriptions of the 10 worst performing classes can effectively boost recognition performance.

**Justification For Why Not Higher Score:**

The paper received unanimous negative reviews and there was no author response.
The reviewers raised valid questions that have remained unanswered.

**Justification For Why Not Lower Score:**

N/A

---

### Decision · Program_Chairs · 2024-01-16

Reject